

# *In silico* testing of flavonoids as potential inhibitors of protease and helicase domains of dengue and Zika viruses

Omar Cruz-Arreola[1,2], Abdu Orduña-Diaz[2], Fabiola Domínguez[3],
Julio Reyes-Leyva[1], Verónica Vallejo-Ruiz[1], Lenin Domínguez-Ramírez[4] and
Gerardo Santos-López[1]

[1] Laboratorio de Biología Molecular y Virología, Centro de Investigación Biomédica de Oriente, Instituto Mexicano del Seguro Social, Metepec, Atlixco, PUEBLA, México
[2] Instrumentación Analítica y Biosensores, Centro de Investigación en Biotecnología Aplicada (CIBA), Instituto Politécnico Nacional, Tepetitla de Lardizábal, Tlaxcala, México
[3] Laboratorio de Biotecnología de Productos Naturales, Centro de Investigación Biomédica de Oriente, Instituto Mexicano del Seguro Social, Metepec, Atlixco, Puebla, Mexico
[4] Department of Chemical and Biological Sciences, School of Sciences, Universidad de las Américas Puebla, San Andrés Cholula, Puebla, Mexico

Corresponding authors
Lenin Domínguez-Ramírez,
julio.dominguez@udlap.mx
Gerardo Santos-López,
gerardo.santos.lopez@gmail.com

## ABSTRACT

**Background**. Dengue and Zika are two major vector-borne diseases. Dengue causes up to 25,000 deaths and nearly a 100 million cases worldwide per year, while the incidence of Zika has increased in recent years. Although Zika has been associated to fetal microcephaly and Guillain-Barré syndrome both it and dengue have common clinical symptoms such as severe headache, retroocular pain, muscle and join pain, nausea, vomiting, and rash. Currently, vaccines have been designed and antivirals have been identified for these diseases but there still need for more options for treatment. Our group previously obtained some fractions from medicinal plants that blocked dengue virus (DENV) infection *in vitro*. In the present work, we explored the possible targets by molecular docking a group of molecules contained in the plant fractions against DENV and Zika virus (ZIKV) NS3-helicase (NS3-hel) and NS3-protease (NS3-pro) structures. Finally, the best ligands were evaluated by molecular dynamic simulations.
**Methods**. To establish if these molecules could act as wide spectrum inhibitors, we used structures from four DENV serotypes and from ZIKV. ADFR 1.2 rc1 software was used for docking analysis; subsequently molecular dynamics analysis was carried out using AMBER20.
**Results**. Docking suggested that 3,5-dicaffeoylquinic acid (DCA01), quercetin 3-rutinoside (QNR05) and quercetin 3,7-diglucoside (QND10) can tightly bind to both NS3-hel and NS3-pro. However, after a molecular dynamics analysis, tight binding was not maintained for NS3-hel. In contrast, NS3-pro from two dengue serotypes, DENV3 and DENV4, retained both QNR05 and QND10 which converged near the catalytic site. After the molecular dynamics analysis, both ligands presented a stable trajectory over time, in contrast to DCA01. These findings allowed us to work on the design of a molecule called MOD10, using the QND10 skeleton to improve the interaction in the active site of the NS3-pro domain, which was verified through molecular dynamics simulation, turning out to be better than QNR05 and QND10, both in interaction and in the trajectory.

**Discussion**. Our results suggests that NS3-hel RNA empty binding site is not a good target for drug design as the binding site located through docking is too big. However, our results indicate that QNR05 and QND10 could block NS3-pro activity in DENV and ZIKV. In the interaction with these molecules, the sub-pocket-2 remained unoccupied in NS3-pro, leaving opportunity for improvement and drug design using the quercetin scaffold. The analysis of the NS3-pro in complex with MOD10 show a molecule that exerts contact with sub-pockets S1, S1', S2 and S3, increasing its affinity and apparent stability on NS3-pro.

# INTRODUCTION

Dengue and Zika are viral diseases transmitted by mosquitoes of the genus *Aedes* (*Petersen, Jamieson & Honein, 2016*; *Wilder-Smith et al., 2019*). Infections occur in tropical and subtropical regions and are considered endemic in more than 124 countries. The most affected regions are the Americas, the Western Pacific and Southeast Asia (*WHO, 2019*). According to the World Health Organization (*WHO, 2018*), about 390 million people are at risk of dengue due to the transmitting vector near them. There are reports of up to 96 million dengue cases with clinical manifestations, causing up to 25,000 deaths per year (*Anwar et al., 2019*; *Petersen, Jamieson & Honein, 2016*; *Wilder-Smith et al., 2019*). Zika has a lower incidence yet represents an important health problem due to its increase since 2015 (*Borchering et al., 2019*). Zika outbreaks in 2007 and subsequently in 2013–2014 (*Ebranati et al., 2019*; *Musso, Nilles & Cao-Lormeau, 2014*) were sporadic and in most cases associated with minor symptoms (*Brasil et al., 2016*; *Sakkas, Economou & Papadopoulou, 2016*). Yet in 2015, a new Zika epidemic emerged in the Americas; and for the first time the alarming presence of fetal microcephaly in the offspring of infected pregnant mothers was noted (*Goh et al., 2019*; *Mlakar et al., 2016*).

Both dengue virus (DENV) and Zika virus (ZIKV) fall in the genus *Flavivirus* of the family *Flaviviridae* (*Mayer, Tesh & Vasilakis, 2017*). Flaviviruses are enveloped viruses containing a ~11 kb genome of positive single-stranded RNA (ssRNA+) which encodes for a ~360 kDa polyprotein. Its proteolytic maturation yields three structural proteins (C, prM and E) and seven nonstructural proteins (NS1, NS2A, NS2B, NS3, NS4A, NS4B and NS5) (*Apte-Sengupta, Sirohi & Kuhn, 2014*; *Perera & Kuhn, 2008*). Four serotypes of DENV have been identified (DENV1, DENV2, DENV3, and DENV4) all of which can cause dengue symptoms and be fatal (*Chin-Inmanu et al., 2019*; *Tsang et al., 2019*). In turn, the known ZIKV belong to a single serotype in which two lineages are distinguished (African and Asian) sharing more than 95% of their amino acid (aa) sequence (*Dowd et al., 2016*). Non-structural proteins (NS) have key roles in the infectious cycle (*Diamond & Pierson, 2015*). NS3 is a multidomain protein and has been identified as a target for drug development (*Othman et al., 2017*; *Timiri, Sinha & Jayaprakash, 2016*). Flaviviral

protein NS3 consists of 618 residues. Its 180 N-terminal residues comprise a region with proteolytic activity (NS3-pro) which participates in the polyprotein maturation and the release of proteins into the cytoplasm (C, NS2A, NS3 and NS5) (*Constant et al., 2018*). At the C-terminus, NS3 residues 181-618, have helicase activity (NS3-hel), responsible for the RNA unwinding that facilitates viral RNA replication through NS5 (*Apte-Sengupta, Sirohi & Kuhn, 2014*; *Lescar et al., 2018*).

Both diseases represent serious global health and economic threats due to their rising incidence and spread to new geographical areas (*Cattarino et al., 2020*; *Gabriel, Alencar & Miraglia, 2019*). There is currently no vaccine or specific antivirals available for them (*Fibriansah & Lok, 2016*; *Poland et al., 2018*; *Poland, Ovsyannikova & Kennedy, 2019*) and this is an active area of development. We have experimentally identified several compounds in purified medicinal plant fractions that inhibit DENV infection (*Flores-Ocelotl et al., 2018*). Some of these compounds, such as quercetin, luteolin, caffeic acid, kaempferol, coumaroyl or their derivatives, have been recognized for exhibiting against DENV activity *in vitro*, as determined in a DENV2 replication assay carried out by group (*Flores-Ocelotl et al., 2018*) and others (*Othman et al., 2017*; *Sahoo et al., 2016*; *Wadood et al., 2017*). It is unknow if these compounds target one or more viral or host pathways. NS3 has been suggested as a target to inhibit the flaviviral cycle.

Computational resources have acquired great value in drug discovery since they can be used to model the ability of molecules to interact with target proteins mimicking biological conditions. Computational analyses add high throughput to the search for drugs and can be easily broadened to related viruses. This process can be useful both against known viruses and to prepare tools to target pathogens as they arise (*Anusuya & Gromiha, 2017*; *Marquez-Dominguez et al., 2020*; *Senthilvel et al., 2013*). The aim of this paper was to use molecular docking and molecular mechanics to identify compounds capable of interacting with either the protease or helicase domains of the NS3 protein of two important flaviviruses (DENV and ZIKV) and thus predict the potential inhibition of their enzymatic activity. For this purpose, we first conducted a comparative structural and identity analysis of the NS3-hel and NS3-pro of both viruses. We show that previously tested plant extracts, particularly quercetin-derivatives, offer a new lead on wide spectrum drug design targeting NS3-proteases.

## MATERIALS & METHODS

### Identity analysis for the DENV and ZIKV polyprotein

Sequence alignments of polyproteins from DENV and ZIKV, and NS3 proteins were carried out using Clustal Omega (https://www.ebi.ac.uk/Tools/msa/clustalo/) (*Mount, 2009*). Viral sequences used for all identity analyses and to calculate the similarity matrix were obtained from the NCBI "Virus Variation Resource" database (https://www.ncbi.nlm.nih.gov/genome/viruses/variation/). We aimed to include up to six viral genomes for ZIKV and six for each DENV serotype from each world region: Africa, Asia, North America, Oceania, South America, and Europe. The selection was made considering recent complete sequences known to infect humans. In the case of ZIKV there were six

genome sequences for each region, but for DENV fewer sequences were available: sequences for DENV3 and 4 were lacking in Africa as well as for all the serotypes in Europe (Table S1). Complete NS2B sequences were also analyzed for identity but are not shown.

### Structural similarity analysis of the NS3 protein in DENV and ZIKV

To compare the 3D structure of the NS3 protein for DENV and ZIKV, the structures for ZIKV (PDBIDs for NS3-pro: 5YOD and for NS3-hel: 5TGX) and for each of the four DENV serotypes (PDBIDs for NS3-pro: 3L6P, 2FOM, 3U1I, 5YVV; for NS3-hel: 2BMF and 2JLS, no structures available for DENV1 and 3 NS3-hel) were obtained from the Protein Data Bank (PDB) (https://www.rcsb.org/) (Table S2). Superposition analysis was performed using ProFit v3.3 (*Sumathi et al., 2006*), first on the complete NS3 proteins and afterwards limited to the active sites: the protease and the ssRNA binding site of the helicase. Root-mean-square deviation (RMSD) expressed in angstroms (Å) for N, C, C $\alpha$ and O atoms of the protein backbone was calculated.

### Selection and preparation of ligands and target molecules

A total of 15 molecules previously identified by our group in extracts of *Taraxacum officinale* and *Urtica dioica* (*Flores-Ocelotl et al., 2018*) were used for the screen (Table 1). These were expanded to 20 since mass spectrometry cannot distinguish between isomers. Their 3D structures were obtained from the PubChem database in SFD format and converted to mol2 format using Avogadro (*Hanwell et al., 2012*). Ligands were prepared prior to molecular docking, by structural optimization using the general AMBER force field (GAFF) to optimize drug geometries (*He et al., 2020*). As target receptors, the best quality crystal structures of NS3 proteins were chosen, to represent each DENV serotype and ZIKV (*Cozzini et al., 2008*; *Scapin, 2006*). The receptors structures in PDB format were loaded to VEGA ZZ version 3.2.0 (obtained from https://www.ddl.unimi.it/cms/index.php?Software_projects:VEGA_ZZ:Download). VEGA ZZ was then used to clean the structures by removing solvent molecules, salts, and ligands. Then, the structures were optimized, and partial charge assignment was done prior to docking.

Two ligands were added as negative and positive controls, respectively: dibasic phosphate and Bz-NIle-KRR-AMC. Phosphate is often used for crystallization and may even be solved in the final structure due to the high concentrations used. Thus, it is expected to be an unspecific binder which can be used as a substrate to measure enzyme kinetics for the DENV protease and is expected to have significant affinity.

### Molecular docking

For docking, ADFR 1.2 rc1 (*Ravindranath et al., 2015*) was used. Ligands and receptors were converted to pdbqt using the scripts included in ADFR. All receptors were aligned to have the same reference coordinates. The grid for docking (trg files) was prepared using agfrgui and the bound peptide from structure 3U1I, as a reference for the binding site location. For helicases, we selected the biggest binding site identified by agfrgui, coordinates −16.525, 44.133, 23.142 and length 31.5, 42.5, 40.5. Boxes for the proteases were created with coordinates −35.132, −28.372, −24.665 and length 31.250, 26.750, 31.250 (spacing

**Table 1**  PubChem accession code of ligand compounds previously identified in *Taraxacum officinale* and *Urtica dioica*.

| Molecule code | Molecule name | Substance SID |
|---|---|---|
| DCA01 | 3,5-Dicaffeoylquinic Acid | 6474310 |
| CAA02 | Caffeic Acid | 689043 |
| CFA03 | 5-O-Caffeoylquinic Acid | 5280633 |
| LNG04 | Luteolin-7-O-Glucoside | 5280637 |
| QNR05 | Quercetin 3-rutinoside | 5280805 |
| CHA06 | Chicoric Acid | 5281764 |
| QND07 | Quercetin 3,4′-Diglucoside | 5320835 |
| KFN08 | Kaempferitrin | 5486199 |
| DCA09 | 4,5-Di-O-Caffeoylquinic Acid | 6474309 |
| QND10 | Quercetin 3,7-Diglucoside | 10121947 |
| QND11 | Quercetin 3-diglucoside | 10211337 |
| QND12 | Quercetin 7,4′-Diglucoside | 11968881 |
| LNR13 | Luteolin-7-O-Rutinoside | 14032966 |
| QND14 | Quercetin 3,5-O-Diglucoside | 44229098 |
| LND15 | Luteolin 7,3′-Diglucoside | 44258089 |
| LND16 | Luteolin 7,4′-Diglucoside | 44258093 |
| LND17 | Luteolin 3′,4′-Diglucoside | 44258099 |
| QND18 | Quercetin 3,3′-Diglucoside | 44259153 |
| KFG19 | Kaempferol-3-Glucoside | 5282102 |
| QNG20 | Quercetin 3-Galactoside | 5281643 |

for both was 0.375). For NS3-pro docking NS2B polypeptides were included. For each ligand, 500 runs with 2.5 million energy evaluations were carried out. This set up keeps the receptor rigid but ligands are flexible. Results were visualized on UCSF Chimera (https://www.cgl.ucsf.edu/chimera/download.html) (*Pettersen et al., 2004*). The results were selected based on largest cluster size first and then by highest affinity to finally graph them.

**Single point MMGBSA after short molecular dynamics**

After docking with ADFR, UCSF DOCK (http://dock.compbio.ucsf.edu/DOCK_6/index.htm) was used to sample different protein-ligand conformations in a set up that lets the receptor move. (*Graves et al., 2008*). The most populated ligand pose from ADFR was selected as the initial conformation for molecular mechanics. Briefly, ligand was energy minimized against a rigid receptor then both were parametrized with AMBER forcefield. Then, a short (*i.e.*, 10000 steps) molecular dynamic was run allowing for the movement of all the atoms in the complex. This was followed by 500 steps of energy minimization and MMGBSA evaluation of the energetics.

**Molecular dynamics and trajectory MMGBSA calculations**

Single point MMGBSA is ran after a short molecular dynamic simulation, usually a few thousand steps. Thus, they are unable to catch ligand dissociation or any change that takes significant evolution time, *i.e.* hundreds of nanoseconds. To try and observe those events we used AMBER20 to run molecular dynamics and MMGBSA calculations over

tens of nanoseconds. The amber19SB forcefield was used for the proteins, OPC model for water molecules, and gaff2 for ligands. Charges for ligands were derived using *ab initio* QM (mp2/6-311g(d)). Neutralizing ions were added to a concentration of 0.15 M, water box was built with a10 angstroms distance from the solute to the box edge. Production simulations were run for 100 nanoseconds (ns) with five repeats. MMGBSA calculations were run with the MMPBSA.py module, using the GB method 8 (*mbondi3*), and a salt concentration of 0.15 M. For negative control we used dibasic phosphate and for a positive control the inhibitor aprotinin as its affinity is known to be in the nanomolar range.

## RESULTS

### Sequence similarity between DENV, ZIKV polyproteins and NS3

It is unclear if the sequence similarity between DENV and ZIKV is enough to allow a single molecule to target both pathogens. To explore this, we conducted sequence analyses. The four DENV serotypes shared 68–78% identity throughout their polyprotein sequence (3,430 aa), while comparison with ZIKV showed lower identity of ~55% (Table S3).

On NS3, DENV serotypes shared 75–86% identity, while the identity was again lower (64–68%) between DENVs and ZIKV (Table S4). NS3-hel (438 residues) was more conserved between DENVs and ZIKV (69–72% identity) (Table S5), than NS3-pro, (180 aa; 51–58% identity) (Table S6).

### Sequence and structure similarity of the NS3-hel from DENV and ZIKV

The RNA binding cleft in the NS3-hel, where the RNA substrate makes temporary contact during unwinding for replication, is made up of two alpha helices from domain II and two alpha helices from domain III (Fig. 1A) (*Xu et al., 2005*). Between DENVs and ZIKV, helices $\alpha7''$ DIII, $\alpha3''$ DIII, $\alpha1'$DII and $\alpha2'$DII had 69–84%, 60–73%, 53–66% and 45–81% identity, respectively and showed higher similarity within DENV serotypes (Table S7).

Superposition of the two DENV and one ZIKV helicase structures available (Table S2) also suggested more similarity between DENV serotypes than to ZIKV. Besides, dengue structures displayed a more closed conformation. Main differences are located at two alpha helices ($\alpha1'$DII and $\alpha2'$DII) that make the RNA entrance to the cavity, with residues D290 and R538 (Fig. 1A, in sticks) in different orientation. The RMSD was 0.8 Åbetween DENV2 and DENV4; 1.1 Åbetween DENV2-ZIKV, and 1.25 Åbetween DENV4-ZIKV (Table S8). Given the similarity in secondary and tertiary structures, we considered NS3-hel as a relevant target for docking in the next part of the study.

### Sequence and structure similarity of the NS3-pro from DENV and ZIKV

The protease carries out an important step in viral maturation: processing the polyprotein into active single proteins. Sequence similarity within DENV serotype was above 93% except for DENV2 that showed 84%. Across dengue viruses, identity was as low as 62% (DENV1 vs DENV4) and as high as 76% (DENV1 vs DENV3). When comparing DENVs to ZIKV, identities dropped to ~55% (Table S6).

Next, we analyzed NS3-pro binding site sequence conservation. We selected each of the catalytic-triad residues (H51, D75 and S135, numbering from structure 2FOM) in a block
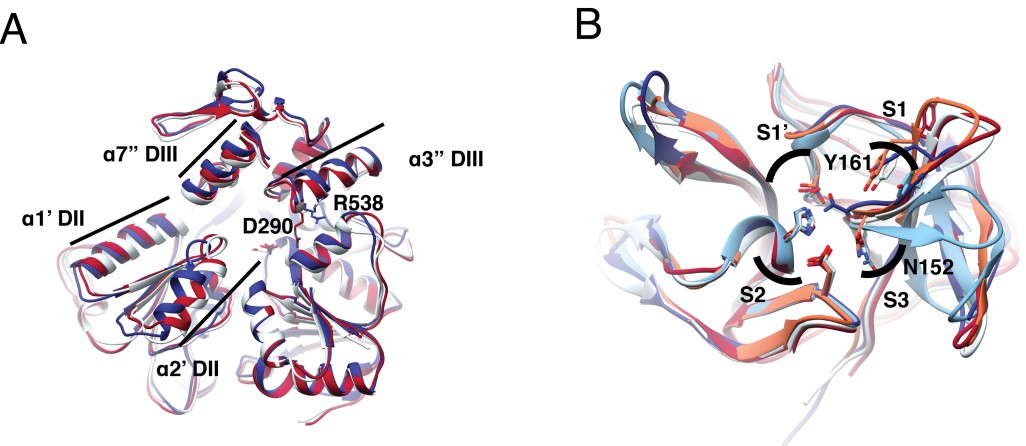

**Figure 1 Structural superposition analysis of NS3-hel (A) and NS3-pro (B) for DENV serotypes and ZIKV.** (A) NS3 alpha helices are labeled according to *Xu et al. (2005)*. Color codes for NS3-hel are DENV2: 2BLF (tan), DENV4: 2JLS (blue) and, ZIKV: 5TXG (magenta). No structures were available for the other DENV serotypes. (B) Superposition of NS3-pro highlighting catalytic residues H51, D75, and S135 (in S1′ and S2 sub pockets) and binding site residues N152 (in S3) and Y161(in S1). Color codes for NS3-prot are DENV1: 3L6P (salmon), DENV2: 2FOM, DENV3: 3U1I (magenta), DENV4: 5YVV (tan), and ZIKV: 5YOD (cyan). Images created in Chimera 1.14 (*Pettersen et al., 2004*).

of 11 aa that define their immediate environment: catalytic residue plus 5 adjacent residues towards the amino- and 5 towards the carboxy-terminus. The block surrounding H51 was 90.90–100% conserved among DENV1, DENV2 serotypes and ZIKV, with 80.81–90.90% when comparing to ZIKV. For the block around D75, identities between DENV serotypes were 54.54–90.90% and 45–63.63% with ZIKV; while for the region around S135, DENV serotypes were 81.81–100% identical and 54.54–72.72% with ZIKV. Thus, the region around H51 was the most conserved (Table S9). Overall, the catalytic site showed high variation in terms of sequence, except for the catalytic triad.

To quantify differences in three dimensions, we analyzed five NS3-pro structures: one for each DENV serotype plus one for ZIKV (Table S2). The catalytic triad did not display much orientation variation and the main difference was observed in the loop with residues Y161 and N152, representing different catalytically important structural conformations in the different proteases (Fig. 1B, sticks). Within DENV serotypes, the structures with the greatest overlap were DENV1-DENV3 with an RMSD of 0.43 Å, followed by DENV1-DENV2, DENV1-DENV4 and DENV2-DENV4 with 0.64 Å, 0.79 Åand 0.83 Å. ZIKV was more like DENV3 with RMSD 0.51 Å, followed by DENV2, DENV1 and DENV4, with 0.63 Å, 0.67 Åand 0.74 Å, respectively (Table S10). Taken together, these data suggest that NS3-pro are diverse in sequence yet conserved in structure, making plausible the goal of identifying a single lead for drug discovery that can target several flaviviruses.

## Molecular docking of ligands against the NS3-hel and -pro domains of DENV and ZIKV

Binding energies for the 20 ligands listed in Table 1 were determined using ADFR and were worked with each of the structures used for 3D superposition. While it is common to use

the best pose, we chose to first rank results using the largest cluster, to focus on the most frequent results and to avoid bias towards infrequent poses with high affinity (Fig. 2).

## NS3-helicase

Despite the different openings of the RNA-binding sites, docking showed similar overall distribution of the 20 ligands in the three NS3-hel structures tested (DENV2, DENV4 and ZIKV) with a mean binding energy of ∼8.3 kcal/mol. Five ligands stood out with affinities <1 standard deviation from the mean, at around -10 kcal/mol: 3,5-dicaffeoylquinic acid (DCA01), quercetin 3-rutinoside (QNR05), Quercetin 3,7-diglucoside (QND10), Quercetin 7,4′-Diglucoside (QND12) and luteolin 3′, 4′-diglucoside (LND17). The first one (DCA01; in Fig. 2A) binds with similar affinity to all the helicases compared. The remaining 4 bind strongly to a different pair of receptors: QNR05 (asterisk) binds tightly to NS3-hel DENV4 and less so to ZIKV, QND10 (+plus sign) binds well to the DENV2 structure at −10.8 kcal/mol while for DENV4 and ZIKV was −8 and −8.7 kcal/mol respectively, QND12 (x cross) binds better to NS3-hel DENV4 and ZIKV, while LND17 (slanted square) binds well to NS3-hel DENV2 and DENV4. However, we observed that there was no binding specificity within the RNA binding site, rather they presented a favorable binding energy but at significantly different locations within the site. This can be observed for DCA01 (Figs. 3A–3C) for QND12 (Figs. 3D–3F) and the rest of the ligands.

After single point MMGBSA (Fig. S1) the results did not show an improvement: no single ligand was observed as a common high affinity binder located at a specific site in the studied NS3-helicases. This is compatible with the ligands docking all along the RNA-binding cleft in the NS3-helicases (Fig. 3), which is large even in DENV structures that tend to be in a closed conformation. The large NS3-hel cleft likely provides little substrate specificity for our small molecules. Despite the high affinities calculated by ADFR or single point MMGBSA for some of the ligands tested, it is unlikely that these ligands are specific to NS3-hel, due to this observed drawback, an exhaustive analysis of molecular dynamics was not carried out.

## NS3-pro

NS3-pro docking (ADFR) results showed that some receptors bind all the ligands slightly better than others: DENV2 had the weakest mean binding energy (−6.2 kcal/mol), followed by ZIKV and DENV1 (−7.4 kcal/mol), while DENV3 and DENV4 showed the highest mean affinities (−7.8 and −8.3 kcal/mol, respectively). Only on DENV4 NS3-pro, two ligands had results better than −10 kcal/mol: quercetin 3-rutinoside (QNR05) and quercetin 3,7-diglucoside (QND10) (Figs. 2B; 4D and 4F). These results are compatible with the low identities between NS3-pro sequences (Table S6) and the observation that binding site residues in sub-pockets S1 and S3 show a different orientation between the structures analyzed (Fig. 1B). In the results ADFR DENV4 had the best affinities with the mentioned ligands, when considering the analysis with DENV3, DCA01 presented an affinity of −9.7 kcal/mol followed by DENV1 with an affinity of −9.1 kcal/mol; therefore, it was also included in MMGBSA analyzes.

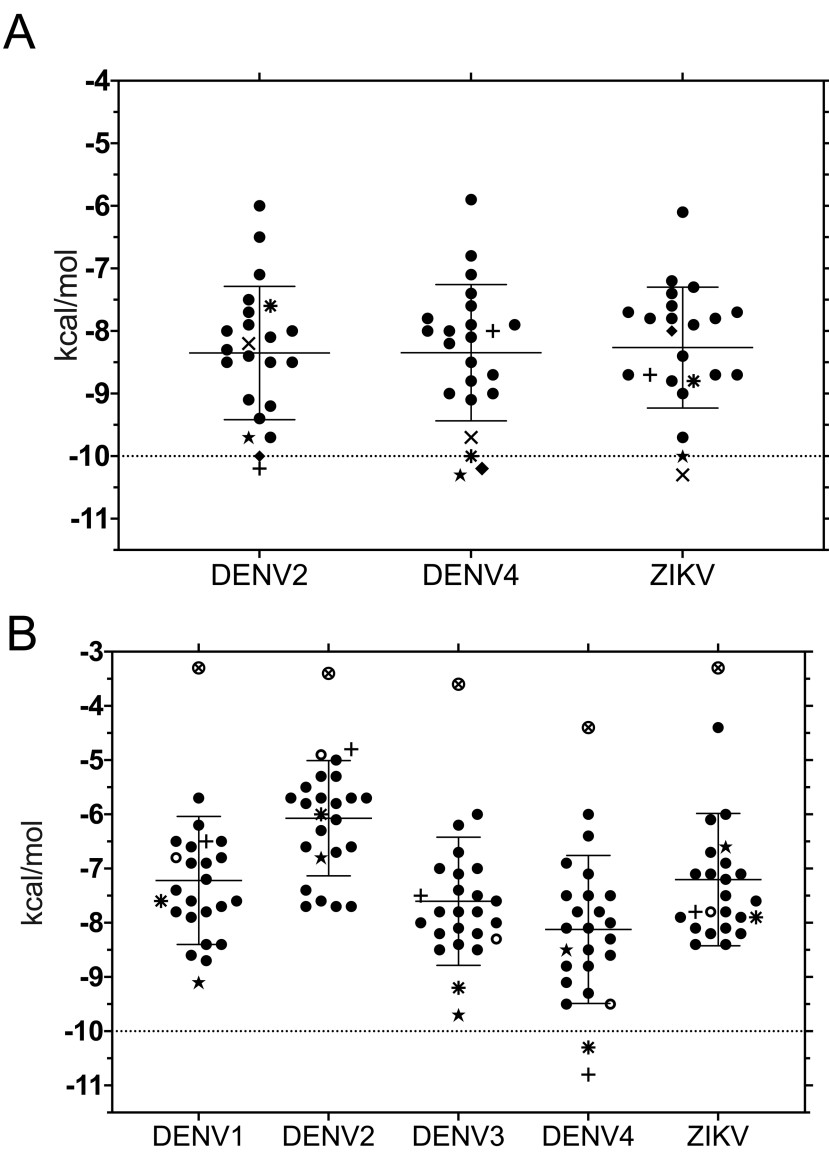

**Figure 2** **Binding affinity (ADFR docking results) in kcal/mol of 20 ligands for three NS3-helicase (A) and five NS3-protease (B) structures.** (A) All helicases have similar profiles and similar mean binding energy for DENV1, DENV3 and ZIKV. Few ligands show high affinity: DCA01 (★), QNR05 (∗), QND10 (+), QND12 (×) and LND17 (♦). (B) Proteases have different profiles and different mean affinities. Few ligands show high affinity: DCA01 (★), QNR05 (∗) and QND10 (+). Dashed lines placed at −10 kcal/mol were considered as an optimal interaction energy value in this study. Plots were obtained using GraphPad Prism v. 8. Dibasic phosphate (⊗) and Bz-NIle-KRR-AMC (○) as negative and positive controls, respectively.

After single point MMGBSA (Fig. S2), DCA01 was found distant from the catalytic residues both in DENV3 and DENV4 NS3-pro (Fig. 4A–4B), occupying part of sub-pocket S1 and S3; in summary the results did not improve for DCA01.

QNR05 and QND10 converged between Y161 and N152 and near S135 (Figs. 4C through 4F) filling up sub-pockets S1′, S1 and S3. In particular, the quercetin core (bold bonds

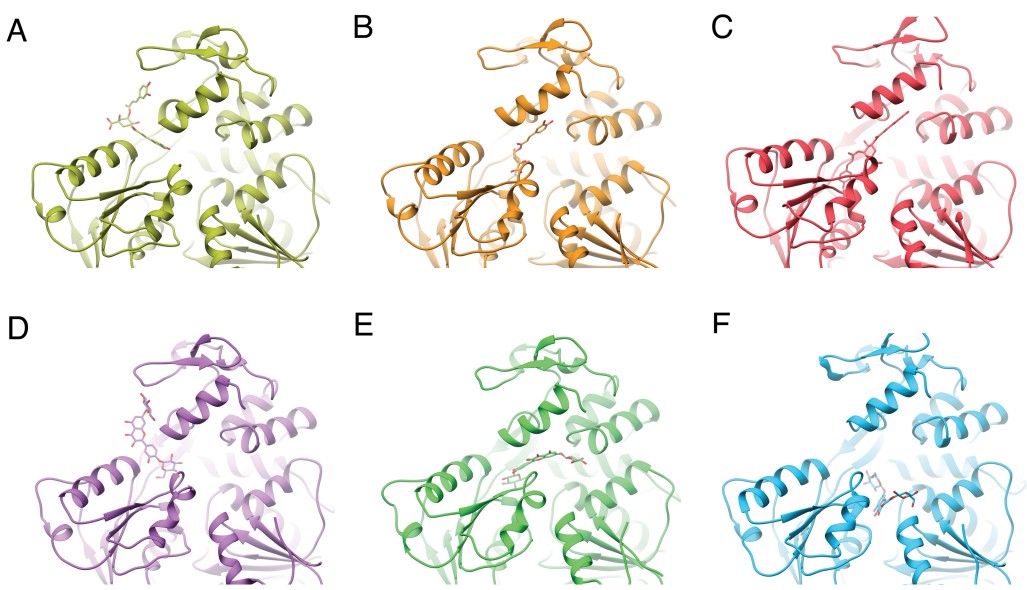

**Figure 3** **Varied binding positions of DCA01 or QND12 (D–F on the different NS3 helicases tested.** DENV2 (A, D) DENV4 (B, E) and ZIKV (C, F), after MMGBSA. Only in DENV2 NS3-hel did ligands bind in between helix $\alpha7''$ DIII and $\alpha1'$DII, just outside the RNA binding site. Images created in Chimera 1.14 (*Pettersen et al., 2004*).

in Figs. 5A and 5B) lodged in sub-pocket S1 in a similar orientation in both DENV3 and DEVN4 NS3-pro, while the saccharides anchored the ligands to the other sub-pockets. Sub-pocket S2 was not visited by any of the ligands. With this analysis we determine that QNR05 and QND10 was the best binder for both DENV3 and DENV4 NS3-pro with a binding energy of −34.41 and −37.05 kcal/mol for QNR05 and −29.86, −36.26 kcal/mol for QND10 and at the same time opens the opportunity to make improvements to these molecules

## NS3-pro molecular dynamics and MMGBSA

The complexes obtained after docking were used as starting points for standard molecular dynamics simulation. Five production simulations were run for each receptor–ligand pair for a time of 100 ns each: a total of four microseconds (µs) simulation time. During the simulations, no ligand remained static relative to the receptor. By examining the evolution of binding energy in time a few trends became apparent. DCA01 is the ligand with the least negative binding energy (Fig. S3). During some simulations the ligand left the receptor completely as the energies drop to 0 (Figs. S3A and S3B).

From the point of view of the receptor and the RMSD of its backbone atoms: all ligands allow for a relatively low for DENV3 (Figs. S4A to S4C). For DENV4, RMSD in the presence of DCA01 reached up to 4 Å (Fig. S4A). For QNR05, the energies when bound to DENV3 are slightly better than those of DCA01. For DENV4, two trajectories displayed almost −40 kcal/mol but 3 were above −20 for a significant simulation time (Fig. S3C and Fig. S3D). RMSD for these complexes is low for both DENV3 and DENV4 (Fig. S4B).

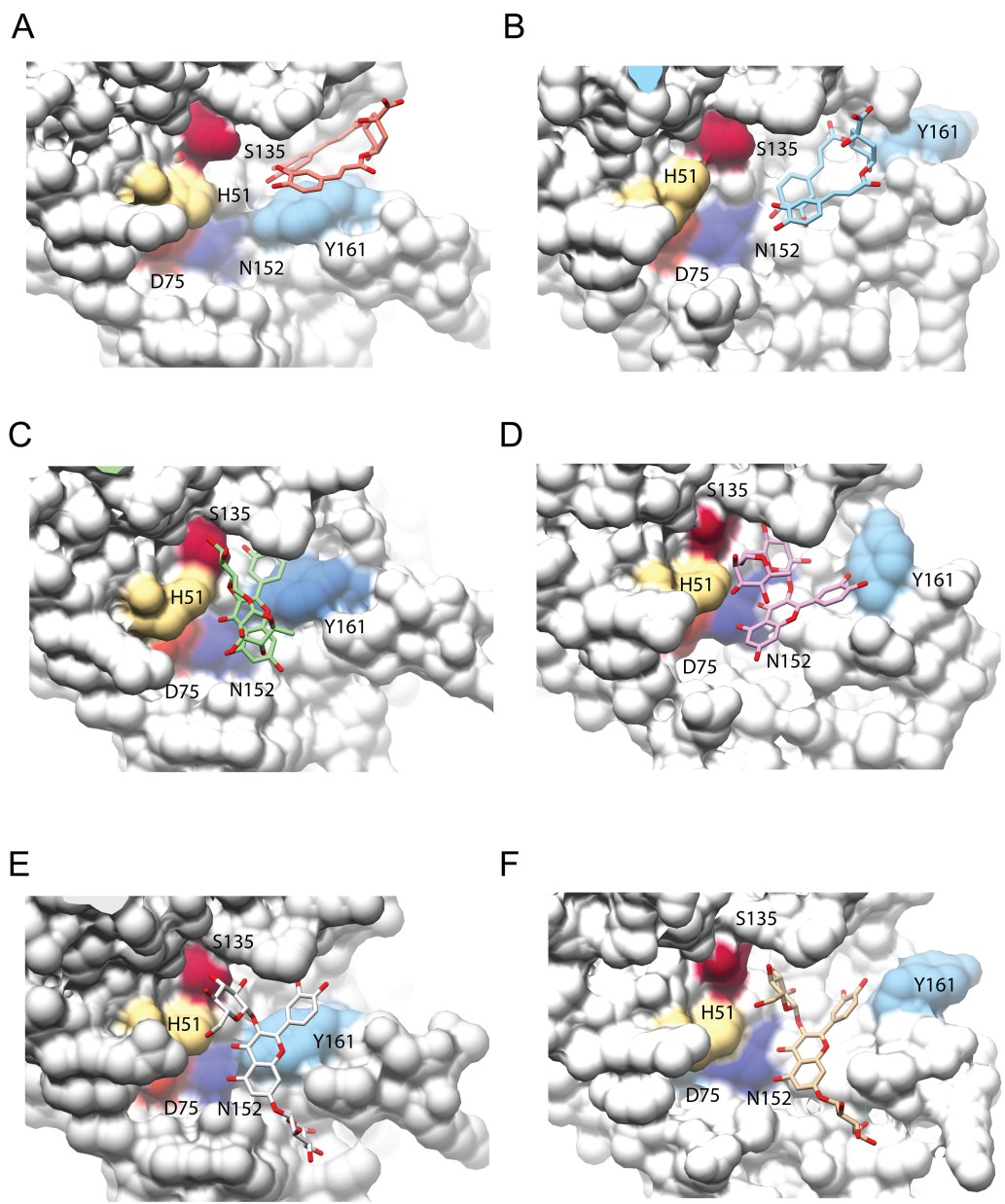

**Figure 4** **Interaction analysis of three ligands with the best binding affinities for the NS3-pro of DENV3 (A, C, E) and DENV4 (B, D, F).** (A–B) DCA01, (C–D) QNR05 and (E–F) QND10. Positions after single point MMGBSA. Residues H, D and S (catalytic triad) and N152, Y161 that form the binding site, are indicated. Images created in Chimera 1.14 (*Pettersen et al., 2004*).

QND10 displayed a more converged evolution: on DENV3 affinities had values around −20 kcal/mol whereas for DENV4 the average was around −40 kcal/mol (Figs. S3E and S3F). RMSD is also low for both DENVs.

Frequency analysis of the hydrogen bonds between NS3-pro and the different ligands (Fig. S6) shows that QND10 shows increased hbonds when compared to DCA01 or QNR05.

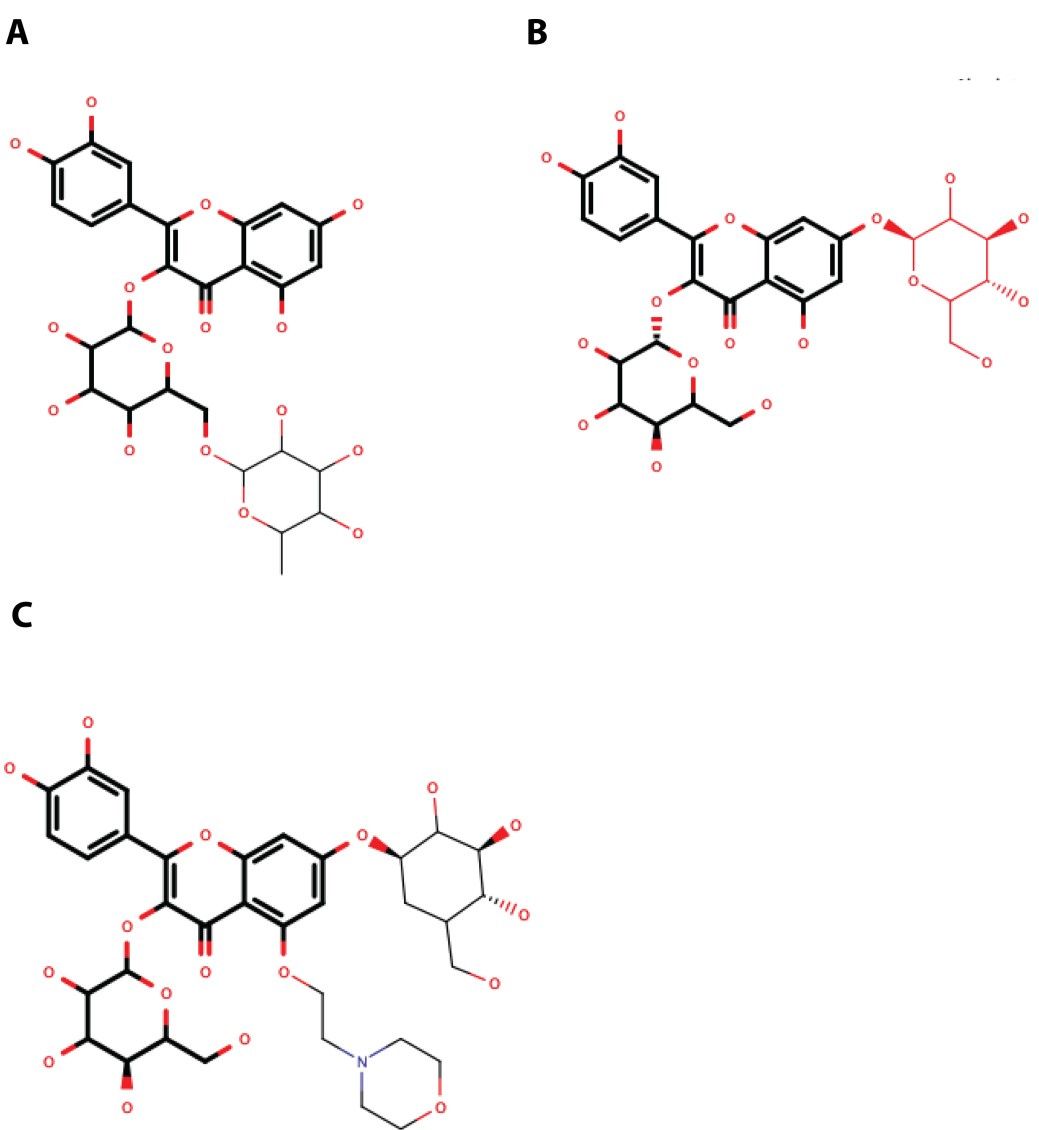

**Figure 5 Structure of the molecules selected as main candidate inhibitors for NS3-pro.** (A) QNR05, (B) QND10 and (C) MOD10 with the common core highlighted in bold. Images created in MarvinSketch.

These results are consistent with the initial docking results yet also put into manifest that the binding site of NS3-pro is a difficult binding site to target. Our ligands bind to the site but display a large apparent $k_{off}$.

Quercetin 3-$\beta$-D-glucoside (CAS no. 482-35-9) is the common core shared by QNR05 and QND10; it anchors these ligands to three of the four sub-pockets. A possible strategy to improve affinity is to use Quercetin 3-$\beta$-D-glucoside as lead. A clear path to do so would be to add a functional group to this core to occupy sub-pocket S2. Our results, guided by the electrostatics of that binding site (*Behnam & Klein, 2020*) (Fig. S5) led us to add a positively charged motif (Fig. 5C). To test this hypothesis, we used the QND10 structure as a template and computationally added a substituent on carbon 6 of the benzene ring.

A new molecule, called MOD10, showed better affinity for both receptor structures, with binding energies of −47.84 kcal/mol for DENV3 and −40.40 kcal/mol for DENV4 (Figs. S3G–S3H). The RMSD values for both targets showed that MOD10 reduced the changes to the structures of DENV3 and DENV4 at around 2 angstroms (Fig. S4D) in contrast with the other more diverse trajectories with the other ligands.

A summary of the MMGBSA energies and comparison to negative and positive controls is shown in Fig. S7.

## DISCUSSION

Flaviviruses are small, enveloped animal viruses containing a single positive-strand genomic RNA. This RNA is transcribed as a polyprotein that undergoes proteolytic processing before releasing mature functional proteins. This virus family comprises pathogens such as dengue, Zika, hepatitis C, West Nile and yellow fever viruses. All of them share the proteolytic processing of a polyprotein into the mature viral proteins crucial for infection (*Klema, Padmanabhan & Choi, 2015*). This activity is carried out by the NS2B/NS3 multifunctional protein; NS3-pro, helicase and RNA triphosphatase activities. Previous experimental *in vitro* work in our group identified 15 compounds that, as two different mixtures, were able to block DENV2 viral replication in BHK21 cell cultures by 75%. The precise viral targets were not identified. In this work, we used NS3-pro and NS3-hel as candidate targets to identify the most likely interactions that are responsible for the inhibition *in vitro*. A secondary question was if one molecule would inhibit targets from multiple viral serotypes.

Sequence similarities for the four DENV serotypes and ZIKV are not that high for both NS3-hel and NS3-pro. But, structurally, the four DENV and ZIKV viruses are rather similar with the NS3-pro catalytic site being the most conserved. These suggest the possibility of a single ligand capable of inhibiting the protease function, at least.

From a binding energetics-point of view, docking to NS3-hel suggested five interesting ligands, in at least two of three NS3-hel structures evaluated. When contrasting the energetics to the binding geometries it becomes apparent that no unique binding site was found and the affinities maybe high but not specific. That is, receptor–ligand recognition is not warranted. The RNA binding site of the NS3-hel is very large due to the need to fit a double stranded RNA molecule. Our results are encouraging but we only tested a binary complex: flavonoid-receptor; we have yet to test a ternary complex: NS3hel-RNA-flavonoid. Molecular dynamics simulations of Zika's NS3hel lend support to a ternary test system for docking (*Mottin et al., 2017*). Further work may benefit from using structures with RNA bound.

NS3-pro presents a different picture: the five receptors analyzed displayed different mean binding energies, likely due to the low conservation of the NS3-pro sequences and the RMSD differences between structures. However, DCA01, QNR05, and QND10 emerged as good ligands for DENV3 and DENV4 NS3-pro. In contrast to what we observed for helicase, results for some the best binders converged at similar 3D positions in these two structures, near the catalytic site, suggesting binding specificity and competitive inhibitory capacity.

Given the limitations inherent to docking methodologies, we followed up on our leads using molecular dynamics. Results revealed that DCA01 readily dissociates from both receptors (DENV3 and DENV4). This is rationalized to the shallowness of the catalytic site and limited degrees of freedom of DCA01 (*Behnam & Klein, 2020*). Although chlorogenic acid, DCA01, and some isomers have shown functional activity *in vitro* against some agents such as Enterovirus and even Hepatitis B virus, they have been through pathways and receptors other than NS3 (*Cao et al., 2017*; *Li et al., 2013*; *Zuo, Tang & Xu, 2015*).

QNR05 and QND10 were observed to bind the sub-pocket S1′, S1 and S3. During molecular dynamics it became clear that, although they remained bound, their positions on the receptor shifted. Their effect was observed on the protein RMSD too, seemingly making them more rigid. We interpret these results, as well as the MMGBSA energies, as indicators that QND10 is a good lead for further drug design. We used this lead as well as the observation that sub-pocket S2 is negatively charged (Fig. S5) to create a new quercetin-derivative, MOD10, that includes a functional group targeting this sub-pocket. It was observed that binding is more stable for MOD10 as well as its effect reducing RMSD changes on the receptors marking an improvement over QND10.

We believe our current results and previous work has identified quercetin derivatives as inhibitors of DENV infection *in vitro* as well as interesting leads to create new NS3-pro inhibitors. However, we cannot confirm that they can inhibit all dengue serotypes. Studies aimed at this objective is what our research group will carry out experimentally in the medium term.

## CONCLUSIONS

Our results suggests that NS3-hel RNA empty binding site is not a good target for drug design. Its conformational changes affect its shape, and it is much bigger that most small molecules. However, there is a secondary RNA binding site worth exploring in later work as well as use of NS3-hel structures with RNA bound. The latter will let us explore a more challenging yet relevant state of this enzyme.

For NS3-pro we have successfully identified drug leads that not only target the catalytic site but have a clear path for improvement through synthesis, or assisted design, using the flavonoid quercetin as the base structure. Our findings suggest that such a new molecule, MOD10, displays improved interaction to NS3pro from DENV3 and DENV4. Further improvements remain to be made. However, the methodology shown here will continue useful to test and evaluate new designs.

## ACKNOWLEDGEMENTS

The authors thankfully acknowledge computer resources, technical advice and support provided by Laboratorio Nacional de Supercómputo del Sureste de México (LNS), a member of the CONACYT national laboratories, with project No. 2018030022C. Katia Hartree Hope is acknowledged for advice on computation time management.

### Funding

Omar Cruz-Arreola had a scholarship from CONACYT (No. 576702) and complementary support from IMSS (97221303). The Consejo de Ciencia y Tecnología del Estado de Puebla (CONCYTEP) Mexico supported the publication of this study. The funders had no role in study design, data collection and analysis, decision to publish, or preparation of the manuscript.

### Grant Disclosures

The following grant information was disclosed by the authors:
CONACYT: 576702.
IMSS: 97221303.
Consejo de Ciencia y Tecnología del Estado de Puebla (CONCYTEP) Mexico.

### Competing Interests

The authors declare there are no competing interests.

### Author Contributions

- Omar Cruz-Arreola performed the experiments, analyzed the data, authored or reviewed drafts of the article, and approved the final draft.
- Abdu Orduña-Diaz performed the experiments, prepared figures and/or tables, conceptualization and Project administration, and approved the final draft.
- Fabiola Domínguez conceived and designed the experiments, analyzed the data, authored or reviewed drafts of the article, conceptualization, and approved the final draft.
- Julio Reyes-Leyva conceived and designed the experiments, prepared figures and/or tables, project administration, and approved the final draft.
- Verónica Vallejo-Ruiz analyzed the data, prepared figures and/or tables, conceptualization, and approved the final draft.
- Lenin Domínguez-Ramírez performed the experiments, analyzed the data, authored or reviewed drafts of the article, conceptualization, and approved the final draft.
- Gerardo Santos-López conceived and designed the experiments, performed the experiments, analyzed the data, prepared figures and/or tables, authored or reviewed drafts of the article, conceptualization and Project administration, and approved the final draft.

### Data Availability

The binding energy data is available in the Supplementary Files.

The raw data resulting from molecular docking is available at figshare: Dominguez-Ramirez, Lenin; Santos-Lopez, Gerardo; Cruz-Arreola, Omar (2022): Dataset for publication. figshare. Dataset. https://doi.org/10.6084/m9.figshare.15127935.v1.

The raw supplemental dynamics data is available at figshare: Dominguez-Ramirez, Lenin; Santos-Lopez, Gerardo; Cruz-Arreola, Omar (2022): Dengue MD files 2021. figshare. Dataset. https://doi.org/10.6084/m9.figshare.19067723.v1.

## Supplemental Information

Supplemental information for this article can be found online at http://dx.doi.org/10.7717/peerj.13650#supplemental-information.

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
