# Peer review of "In silico testing of flavonoids as potential inhibitors of protease and helicase domains of dengue and Zika viruses"

_PeerJ, doi:10.7717/peerj.13650_

## Round 0.1 · original submission · Major Revisions

The reviewers raise several issues that should be addressed. Please revise your manuscript accordingly.

Reviewer 1 asks for experimental validation of your results. In my opinion, additional experimental results are not necessary for publication, but would of course strengthen the manuscript.

In addition to the reviewer comments, I'd like to point out that the Abstract doesn't seem fully aligned with the Conclusions paragraph. Conclusions start with "Our results suggests that NS3-hel RNA empty binding site is not a good target for drug design." This looks like a major finding to me, but it's not mentioned in the abstract as far as I can tell. Please think carefully what your main conclusions are, and then rewrite the Abstract or the Conclusions or both such that they are consistent and report a single message.

Reviewer 1 ·

Basic reporting

no comment

Experimental design

no comment

Validity of the findings

no comment

Additional comments

This well written manuscript describes the in silico based discovery of some compunds with activity against the dengue and zika viruses proteins.

While the computational data certainly looks promising, the authors did not demonstrate biological efficacy of any of the discovered molecules.

To be considered for publication, I recommend that the authors demonstrate antiviral efficacy of at least one of the identified compounds in a cell based system. Dengue and Zika have several excellent model systems which can be handled at BSL-2 and could be used for this.

I cannot recommend this manuscript for publication without at least the proof of concept that the computational approach was able to identify a compound with antiviral activity in a biological system.

·

Basic reporting

minor mistakes

1. line 30: like in the background, please change this statement as some antivirals exist and developed this year. " have been approved"
2. line 35 molecular dynamic simulations
3. line 29: why only microcephaly as there are many public health issues with Zika and Dengue.

Experimental design

the experimental design looks fine.,

Validity of the findings

results look expected
there should be validation of the models like positive and negative control.

Reviewer 3 ·

Basic reporting

The research topic is timely. The results are exposed, and the discussion is written clearly. The literature references are appropriate and correct. However, the quality of the manuscript, including writing quality, figure captions, and table captions, needs to be significantly improved.

Some of the writing issues:
Abstract:
1. Methods. Methods (written twice)
2. Discussion: The analyzes on MOD10? (correct it)

Lines:
36: Methods. Methods (written twice)
144-146: (Structure the sentence correctly)
165-166: populated ligand pose from ADFR was selected as then (the ?) initial conformation.
384: useful to test, a evaluate new. (correct it)


Tables:

In supplemental tables S3 and S4, what are these numbers (percentages?) Write that in the figure caption.

In table S7, figure caption (3rd line): fo domain III (of ?)


Figures:

No figure captions for the supplemental figures. Isn't it very confusing without the figure captions?

Figure S4: What are red and black trajectories in the RMSD vs time graphs? (DENV3 or 4?)

Experimental design

In this study, the authors explored the binding mechanism of various compounds as potential inhibitors of protease and helicase domains of dengue and Zika viruses via molecular docking and MD simulations. Here, the authors used the flavonoids previously shown to inhibit the dengue virus serotype 2 replication in vitro to identify the specific target domains in DENV and ZIKV. Here, the authors chose to explore the NS3-pro and NS3-hel regions for the binding activity of flavonoids. What is the rationale for using the biggest binding sites for helicases? There is no necessary reason for doing this because binding sites of this size are not usually actual binding sites for small molecules.

In the abstract, the author mentioned that molecular dynamics analysis was carried out using UCSF DOCK. Later in the main text, they said that they also used the AMBER20 suite for MD simulations. Can you explain more clearly which one of these you used? If you are used both, why? AMBER20 suite itself is good enough for end-to-end MD simulations.

Validity of the findings

Molecular docking and MD simulations identified that quercetin derivatives could inhibit the DENV infections via interaction with NS3-pro in DENV3 and DENV4 serotypes. These findings were clearly stated, and the discussion was written clearly and concisely. The conclusions are sound.
However, in addition to the MMGBSA calculations, hydrogen bond analysis is recommended for the protein-ligand complex. It would give a better idea of the strength of hydrogen bond interaction.

Additional comments

I support its publication in PeerJ after

1. significantly improving the writing quality, figure, and table captions.
2. hydrogen bond analysis for the protein-ligand complex.

---

## Round 0.2 · accepted · Accept

Thank you for carefully revising your manuscript.